# New Insights into Renal Failure in a Cohort of 317 Patients with Autosomal Dominant Forms of Alport Syndrome: Report of Two Novel Heterozygous Mutations in COL4A3

**DOI:** 10.3390/jcm11164883

**Published:** 2022-08-19

**Authors:** José María García-Aznar, Luis De la Higuera, Lara Besada Cerecedo, Nerea Paz Gandiaga, Ana Isabel Vega, Gema Fernández-Fresnedo, Domingo González-Lamuño

**Affiliations:** 1Healthincode, 15008 A Coruña, Spain; 2Servicio de Genética, Hospital Universitario Marqués de Valdecilla, 39008 Santander, Spain; 3Servicio de Nefrología, Hospital Universitario Marqués de Valdecilla, 39008 Santander, Spain; 4Servicio de Pediatría, Hospital Universitario Marqués de Valdecilla, 39008 Santander, Spain

**Keywords:** Alport syndrome, autosomal dominant inheritance, ESKD, COL4A3, COL4A4

## Abstract

Alport syndrome (AS) is a clinically and genetically heterogeneous disorder with a wide phenotypic spectrum, onset, and progression. X-linked AS (XLAS) and autosomal recessive AS (ARAS) are severe conditions, whereas the severity of autosomal dominant AS (ADAS) may vary from benign familial hematuria to progressive renal disease with extra-renal manifestations. In this study, we collated information from the literature and analyzed a cohort of 317 patients with ADAS carrying heterozygous disease-causing mutations in COL4A3/4 including four patients from two unrelated families who carried two novel variants in COL4A3. Regarding the age of onset of the disease, 80% of patients presented urinalysis alterations (microhematuria, hematuria, and/or proteinuria) before the age of 40 years. The cumulative probability of suffering adverse renal events was mainly observed between 30 and 70 years, without statistical differences between COL4A3 and COL4A4. We observed statistically significant differences between the sexes in the age of developing ESKD in cases affected by mutations in COL4A3/4 (*p* value = 0.0097), suggesting that males begin experiencing earlier deterioration of renal function than women. This study supports the importance of follow-up in young patients who harbor pathogenic mutations in COL4A3/4. We update the knowledge of ADAS, highlighting differences in the progression of the disease between males and females.

## 1. Introduction

Alport syndrome (AS) is a progressive hereditary kidney disease accompanied by sensorineural hearing loss and ocular abnormalities caused by disease-causing mutations in the COL4A3, COL4A4, and COL4A5 genes, which encode type IV collagen α3, α4, and α5 chains, respectively [1]. Monoallelic deleterious variants in COL4A3 and COL4A4 cause autosomal dominant forms of Alport syndrome (AS). This glomerulopathy with a wide range of severity can present autosomal recessive, autosomal dominant, or X-linked inheritance. The most frequent type of AS is caused by loss-of function (LOF) mutations in COL4A5, responsible for X-linked disease, which explains about 80% of AS cases. Autosomal recessive inheritance caused by biallelic mutations in COL4A3/4 is observed in about 15% of cases, whereas less than 5% of cases show autosomal dominant inheritance due to the presence of heterozygous mutations [1,2,3]. Nevertheless, other renal manifestations are caused by LOF mutations in COL4A3/4 in cases misdiagnosed as AS. For instance, in cases of thin basement membrane nephropathy (TBMN), heterozygous mutations in COL4A3/4 are present in 40% [4]; in focal segmental glomerulosclerosis (FSGS), 10% of cases harbor LOF mutations in the mentioned genes [5]; in IgA nephropathy, it has been suggested that individuals with a mutation in the NC1 domain develop antigenic properties against alveolar and glomerular epithelia [6].

In X-linked AS, male patients develop a more severe phenotype than heterozygous females or mosaicism cases. More severe presentations of the disease may result in a lower age of onset and early progression to end-stage renal disease. The phenotypic spectrum of dominant forms may vary from benign familial hematuria to progressive renal disease with extra-renal manifestations. Renal features include microhematuria, hematuria, or proteinuria, with onset of the disease during the second decade of life and variable deterioration of renal function leading to ESKD due to thickening and thinning across the glomerular basement membrane. Extra-renal manifestations may include sensorineural hearing loss and ocular abnormalities, which happen in less than 15% of cases [7]. An estimate of median renal survival time in previously reported reviews of large cohorts of patients was 70 years in dominant forms [2], 50 years later than what had been observed in X-linked and autosomal recessive AS, and the mean age at ESKD was about 55 years old [8,9].

Therefore, the high variability of manifestations in ADAS and the presence of undiagnosed cases make establishing the clinical diagnosis before a genetic test is performed difficult. The inclusion criteria in previous cohorts of ADAS included autosomal dominant familial microhematuria and proteinuria with or without CKD or abnormalities in the glomerular basement membrane [9]. It is worth noting that family history may not necessarily be a criterion, since a few cases have been found to occur de novo. Testing for genetic variants in COL4A3/4 in autosomal dominant AS has been widely recommended by different clinical practice guidelines, with high importance when urinalysis alterations are identified in patients who are at significant risk of renal function deterioration [10].

In this study, we provide the relevant data on disease prognosis in autosomal dominant forms of Alport syndrome in a large cohort of 317 cases including two novel deleterious missense mutations in COL4A3, described in this report. The analysis of the collected clinical data provides new insights for the management of autosomal dominant AS that includes analytical abnormalities that mark the onset of the disease and sex stratification of the renal progression of the disease.

## 2. Materials and Methods

### 2.1. Review of Reported Cases

We collected clinical data on the reported autosomal dominant AS cases with deleterious mutations in COL4A3/4. For this purpose, we used a systematic search of combining keywords by using PubMed resources (“COL4A3”, “COL4A4”, “mutation”, “variant”, and “Alport syndrome”, “renal failure”, “renal insufficiency”, “hematuria”, “ESKD/ESKD/CKD/CRD”) and incorporating them into an internal database. Seventy papers were revised. A total cohort of 317 patients were recruited including four cases studied at our center who harbored two novel likely pathogenic mutations in COL4A3 (Appendix A). The variants selected from patients reported in the screened literature were classified as pathogenic or likely pathogenic according to ACMG criteria. The allele frequency of these variants were lower than 0.01%, except for two variants of uncertain clinical significance in the COL4A4 gene (p.Met1399Leu and p.Gly545Ala), which were considered as low-penetrant polymorphisms. For this study, we designed a specific form to collect the demographic data of index cases and family members, genotype results from genetic tests, and phenotypic features including analytical abnormalities, state of renal function, age of disease onset, and adverse events related to renal failure. Twenty published reports that matched the search criteria and included this detailed clinical information were gathered for statistical analysis.

### 2.2. Patients and Family Description

**Case 1:** A 15-year-old girl who presented at the age of 13 with transient gross hematuria and massive proteinuria, with normal serum electrolytes, albumin, urea, and creatinine values (albumin, 3.2 g/dL; urea, 24 mg/dL; serum creatinine, 0.6 mg/dL; and estimated glomerular filtration rate [eGFR], 106.7 mL/min/1.73). She also had a normal hemoglobin level (12.8 mg/dL), normal white blood cell count (9.3 × 10^3^/µL), and normal platelet count (240 × 10^9^/L). Peripheral blood film showed anisopoikilocytes (irregularly shaped and burr cells), but no leukocyte inclusion bodies. She had no history of hearing difficulties or vision problems, but her mother (43 years old) had a history of hearing difficulties, microscopic hematuria, and history of transient gross hematuria attributed to urinary tract infections, with her eGFR being 80.0 mL/min/1.73. Her grandfather (father of the proband’s mother) had renal failure and was undergoing dialysis at the age of 70. On examination following referral, the patient had no edema, normal blood pressure (93/63 mmHg, at the 50th percentile for age and height), and normal heart rate. Urine examination revealed mild proteinuria, albumin-to-creatinine ratio (ACR) of 140 to 286 mg/g, and hematuria, with serum electrolyte, urea, and creatinine levels normal and eGFR of 96.2 mL/min/1.73. Ultrasonography revealed normal-sized kidneys with normal parenchymal echogenicity but some loss of corticomedullary differentiation. Slit-lamp examination and audiometry were normal. She is on regular follow-up, and her proteinuria is being treated with an angiotensin-converting enzyme inhibitor (ACEI). There was a significant reduction in proteinuria, which ranges from normal through mild to low-moderate increase (ACR, 24 to 100 mg/g), but persistent microscopic hematuria. She did not experience any new episodes of gross hematuria.

**Case 2:** A 31-year-old male who presented at the age of 26 years with corticoresistant nephrotic syndrome (edema, hypoalbuminemia, and proteinuria of >3.5 g/day). Renal biopsy revealed lesions consistent with a diagnosis of focal and segmental hyalinosis. Immunosuppressor treatment with cyclosporine and mycophenolate was initiated, showing bad response. During the next years of follow-up, he has suffered a slow deterioration of renal function, maintaining high levels of proteinuria. He is currently 31 years old and has developed ESKD. A brother of this proband has the same condition.

### 2.3. Informed Consent

For both underage patients studied at our center, their parents provided written informed consent for the genetic tests. The studies also had the authorization of those responsible for the Pediatric Services of the Hospital Universitario Marqués de Valdecilla (Santander, Spain).

### 2.4. Genetic Analysis

The two cases reported in this study were included in a High-Throughput Sequencing protocol and their family members were analyzed following the procedure for Sanger sequencing. DNA was extracted from the blood samples of the two probands and their family members and purified with QIAsymphony SP^®^ (Qiagen). The samples were prepared using the Agilent SureSelect Library Preparation Kit for Illumina paired-end multiplexed sequencing according to the manufacturer’s instructions. Enrichment of regions of interest was performed using a SureSelect Probe Kit (Agilent) that selectively captures the coding regions and adjacent intronic areas of the selected genes of the whole clinical exome. Cluster preparation was carried out using the cBot (Illumina, San Diego, CA, USA) device. Bioinformatics analysis was performed through an end-to-end in-house pipeline developed by Health in Code (A Coruña, Spain), in accordance with the best WES analysis practices. At the interpretation level, we performed variant analysis using a glomerular disease panel that included 124 genes (*ACTN4*, *ADAMTS13*, *ALG1*, *ALMS1*, *ANLN*, *APOA1*, *APOE*, *APOL1*, *ARHGAP24*, *ARHGDIA*, *AVIL*, *B2M*, *C1QA*, *C1QB*, *C1QC*, *C3*, *C4A*, *C4B*, *CASP10*, *CD151, CD2AP, CD46, CD81, CDK20, CFB, CFH, CFHR1, CFHR2, CFHR3, CFHR4, CFHR5, CFI, CFP, COL4A3, COL4A4, COL4A5, COL4A6, COPA, COQ2, COQ6, COQ8B, CRB2, CTLA4, CUBN, DGKE, DLC1, DNASE1, EMP2, FAS, FASLG, FAT1, FCGR2A, FCGR3A, FGA, FN1, G6PD, GATA3, GLA, HAS2, IFT140, INF2, IRF5, ITGA3, ITGAM, ITGB4, ITSN1, KANK1, KANK2, KANK4, LAGE3, LAMB2, LCAT, LMX1B, LYZ, MAFB, MAGI2, MMACHC, MYH9, MYO1E, NEU1, NPHS1, NPHS2, NUP107, NUP133, NUP160, NUP205, NUP85, NUP93, NXF5, OCRL, OSGEP, PAX2, PDSS2, PGM3, PLA2R1, PLCE1, PMM2, PODXL, PRKCD, PTPN22, PTPRO, SCARB2, SEC61A1, SGPL1, SLC7A7, SMARCAL1, STAT1, STAT4, TBC1D8B, THBD, TNFSF4, TNIP1, TNS2, TP53RK, TPRKB, TREX1, TRPC6, TTC21B, VTN, WDR73, WT1, XPO5, ZAP70, ZMPSTE24*).

Sanger sequencing was performed in family members of cases 1 and 2, using a specific primer designed for the region of interest.

### 2.5. Statistical Analysis

The cumulative probability of the occurrence of renal failure including ESKD and chronic renal disease (CRD) and replacement therapy such as kidney transplantation and dialysis was estimated using the Kaplan–Meier method, and factors were compared using the log-rank (Mantel–Cox) method. Survival was calculated from birth. The patients included in the analysis of cumulative probability to suffer adverse events were censored at the time of last follow-up, as mentioned in the publication. A 2-sided *p* value < 0.05 was considered to indicate the statistical significance. The differences in the longitudinal follow-up of each patient from the publications is a technical limitation of this analysis.

## 3. Results

### 3.1. Review of Cases

A summary of the clinical data from a cohort of 317 cases (169 males and 148 females) is shown in Table 1 and Table 2. This cohort included patients previously reported in the scientific literature (313) and four cases studied at our center who harbored pathogenic or likely pathogenic heterozygous mutations in COL4A3 or COL4A4. These data were used to calculate the cumulative probability of the occurrence of renal failure including ESKD and CRD and replacement therapy such us kidney transplantation and dialysis. We registered 136 cases with deleterious mutations in COL4A3 and 181 in COL4A4.

Regarding COL4A3, 94.1% of the mutations were missense, 1.5% in-frame deletions, 0.74% splicing, 0.74% nonsense, and 2.9% frameshifts. For COL4A4, 54.1% were missense, 16.6% splicing, 14.9% frameshift, 9.4% nonsense, and 5% in-frame deletions. In total, non-truncating mutations in COL4A3/4 (non-synonymous and in-frame deletions) were present in 74.8% of the included patients, whereas truncating mutations (nonsense, frameshift, and splicing) were observed in 25.2% of the cases.

Regarding the onset of renal disease, alterations in urinalysis such as microhematuria, hematuria, or proteinuria were indicative of the diagnosis of Alport syndrome with the full phenotypic spectrum of the disease. Eighty percent of the patients with heterozygous deleterious mutations in COL4A3/4 showed some kind of analytical urine alteration before the age of 40 years, with an average age of detection of 22 years in patients with mutations in COL4A3 and 18 years in patients with mutations in COL4A4 (Figure 1).

Adverse events associated with renal function deterioration (ESKD and CRD) and renal replacement therapy (dialysis and renal transplant) occurred in 24.3% and 22.1% of patients with mutated COL4A3 and COL4A4, respectively (Figure 2). The events were accumulated in the age range of 30–70 years, with borderline statistical differences between the sexes (*p* value = 0.03) in cases affected by mutations in COL4A3/4 (Figure 3). Excluding patients with CRD and considering patients with end-stage renal disease, we observed statistically significant differences between the sexes (*p* value = 0.0097). Male patients with mutated COL4A3/4 experienced an earlier age of ESKD than females, approximately five years earlier (35 years in males and 40 years in females). No statistically significant differences were observed in the comparison of the genes (COL4A3 vs. COL4A4).

### 3.2. Novel Missense Mutations in COL4A3

We identified two novel missense mutations in COL4A3, p.Gly315Ser and p.Gly366Arg. These changes substitute one glycine residue, which is an amino acid with an aliphatic hydrophobic side chain, for serine, which is an amino acid with a neutral polar side chain, and arginine, which is an amino acid with a basic side chain. These variants are located in the first glycine of the Gly-X-Y repeats of the triple-helical region in the COL4A3 sequence, which is essential for conserving the structure and stability of heterotrimeric chains of collagen IV.

The first index patient (case 1) of this study harbored p.Gly315Ser. This variant was not found in the gnomAD database [11], and it segregated with the disease in the family (Figure 4), showing full penetrance but variable expression of the phenotypic presentation of renal disease in the mother and maternal grandfather. This fact was in agreement with the wide range of features associated with heterozygous defects in COL4A3/4 in autosomal dominant forms of Alport syndrome. In addition, the index patient carried a rare heterozygous mutation in COL4A3 and p.Arg1661Cys, which is a variant with conflicting interpretations of pathogenicity from VUS to pathogenic. This mutation presents a relatively high frequency in the control population and a lack of cosegregation in the first reported family of this study, being observed in healthy family members (brother of the index case), so we considered that its effect on the protein is likely to be neutral. We also identified the polymorphism deletion CFHR3-CFHR1 in homozygosity, which may be of importance in the case of renal failure and transplantation in the context of susceptibility to atypical hemolytic uremic syndrome (aHUS).

The second index patient (case 2) of this study harbored p.Gly366Arg. This mutation has not been previously reported in the literature, but it has been classified as being likely pathogenic for AS in databases, being observed in three heterozygous carriers in the gnomAD database [11]. This change was also detected in his brother, who was affected by the same condition (Figure 4). The parents could not be evaluated, although we suspect that the variant was inherited from one of the progenitors.

About 350 pathogenic and likely pathogenic mutations have been reported in COL4A3 including missense, nonsense, and frameshift variants as well as deletions and duplications. Focusing on the missense changes, most of them affected a glycine residue in the Gly-X-Y repeats of the triple-helical region. According to ACMG* (American College of Medical Genetics and Genomics) [12], both mutations are considered to be likely pathogenic on the basis of the fulfilled criteria* of the population data (PM2), computational and predictive data (PM5 and PP3), functional data (PM1), segregation data (PP1), and phenotypic data (PP4).


**ACMG Criteria that were met by p.Gly315Ser and p.Gly366Arg:*



*PM2: Absent from controls (or at extremely low frequency if recessive) in the Exome Sequencing Project, 1000 Genomes Project, or Exome Aggregation Consortium.*



*PM5: Novel missense change at an amino acid residue where a different missense change determined to be pathogenic has been seen before.*



*PP3: Multiple lines of computational evidence support a deleterious effect on the gene or gene product (conservation, evolutionary, splicing impact, etc.).*



*PM1: Located in a mutational hot spot and/or critical and well-established functional domain (e.g., active site of an enzyme) without benign variation.*



*PP1: Cosegregation with disease in multiple affected family members in a gene definitively known to cause the disease.*



*PP4: Patient’s phenotype or family history is highly specific for a disease with a single genetic etiology.*


## 4. Discussion

Collagen IV is the most abundant of the structural components of basement membranes of the epithelium. This component is a multimeric protein composed of three subunits of α chain collagen IV that are organized in a triple-helical conformation. This protein has the highest expression in the epithelium of the kidney, lung, glands, and aorta. Each α-chain subunit is encoded by six different genes. In the majority of tissues, except in the smooth muscle, collagen IV protomers are composed of two α1 chains and one α2 chain, whereas α3/α4/α5/α6 chains show a more limited distribution. Mammals have two different nets of collagen IV in the glomerular basement membrane. In the subendothelium, collagen IV is composed of α1/α2 chains and the subepithelial layer contains α3/α4/α5 chains. The α1/α2 chains are synthesized by endothelial cells, mesangial cells, and podocytes, whilst α3/α4/α5 are synthesized exclusively by podocytes [13,14,15]. The specific location of α3/α4/α5 chains suggests different biochemical properties of COL4A3/4/5.

Disease-causing mutations in the six α chains trigger different phenotypes with marked differences according to the affected tissue. Focusing on α3/α4/α5 defects, COL4A3/4 are encoded by autosomal chromosomes, whereas the X-chromosome encodes COL4A5. Recent advances in NGS have enabled genetic testing to be performed for the diagnosis of AS as the first-line diagnosis [16,17], allowing for the analysis of single-nucleotide variants and copy number variation. The genotyping of family members in the context of AS has revealed a wide spectrum of the phenotype including genotype–phenotype correlations. Dominant forms of AS represent up to 30% of AS suspicions [18,19,20] and are characterized by a spectrum of phenotypes ranging from progressive kidney disease with extra-renal abnormalities to isolated hematuria with a non-progressive, or very slow, course with a wide range of expression, inheritance, and penetrance. In this study, we provide already known predictive data related to the onset of the disease and deterioration of renal function, and new insights into renal failure in patients with autosomal dominant forms of AS from a large cohort of patients.

In a cohort of 317 patients with heterozygous deleterious mutations in COL4A3/4, almost 80% of them showed urinalysis alterations (microhematuria, hematuria, and/or proteinuria) before the fourth decade of life. Therefore, although the composition of collagen IV in the glomerular basement membrane may show anomalies in its structure and organization, renal function is not affected during the first decades of life. Urine analysis in family members with identified deleterious mutations in COL4A3/4 results is important in the management and follow-up of the disease, and so is genetic testing, being essential for the identification of subclinical states of the disease in young family members. There is a relatively high incidence of microhematuria and normal renal function in carriers of deleterious mutations in COL4A3/4 [10].

Regarding renal failure, our results showed that adverse events associated with renal function deterioration (ESKD and CRD) and renal replacement therapy (dialysis, renal transplant) were accumulated among the 30–70-year-old patients with deleterious mutations in COL4A3/4. Surprisingly, we found slight differences between the sexes in the cases affected by mutations in COL4A3/4 with a borderline *p* value (*p* = 0.03), and stronger statistically significant differences between the sexes considering ESKD and renal failure and excluding CRD as events (Figure 3). Male patients affected by defects in COL4A3/4 started to suffer adverse events after the age of 35 years, whereas most female patients started exhibiting events after the age of 40 years. The observed differences between the sexes in the context of mutated COL4A3/4 suggest that collagen IV nets may have different properties or susceptibility to genetic defects between males and females. However, another group of authors who conducted a similar analysis of a large cohort of patients with dominant forms of Alport syndrome (COL4A3 and COL4A4) before our study [9] did not observe statistical differences between the sexes. Therefore, these results support the importance of studying the nature of the genetic findings and their comparison with other different sets of variants, whose consequences may entail personalized management of the disease. This work will require additional investigations to identify the underlying factors of these observations.

The unfeasibility of a real birth-time of event/censored follow-up is a bias that needs to be taken into account when interpreting the results of the statistical analysis. This limitation has a more relevant impact for patients from the literature, where some non-published factors (striking phenotype) might also be present.

In addition, we report two novel mutations in COL4A3 that affect the glycine residues (p.Gly315Ser and p.Gly366Arg), one of the main pathogenic mutation types and amino-acid changes in collagen IV genes. The p.Gly315Ser mutation was segregated in four affected family members and was absent in a healthy brother of the proband. On the other hand, the rare variant p.Arg1661Cys in COL4A3 was found in the affected index case and the healthy brother, suggesting that it was present in trans in the proband, likely inherited from the paternal side, without a reported positive family history. These data support that p.Arg1661Cys had a likely neutral role in the disease of this family. The mutation p.Gly366Arg was also predicted to be likely pathogenic, and it was identified in the proband’s brother with the same condition. The presence of heterozygous carriers of variants in COL4A3/4 in the general population could be explained by subclinical forms of ADAS in patients who harbor pathogenic mutations in these genes.

## 5. Conclusions

In conclusion, our results support the importance of follow-up in young patients affected by autosomal dominant AS and the genetic screening of all family members who do not show any clinical symptoms. Male patients affected by COL4A3/4 mutations showed an earlier age of ESKD by approximately 5 years than the females, although these data need to be replicated in other patient cohorts. These results are important for the stratification of patients with these features in order to improve the management of autosomal dominant AS and better understanding the factors that influence the inter- and intra-familial variability of ADAS.

## Figures and Tables

**Figure 1 jcm-11-04883-f001:**
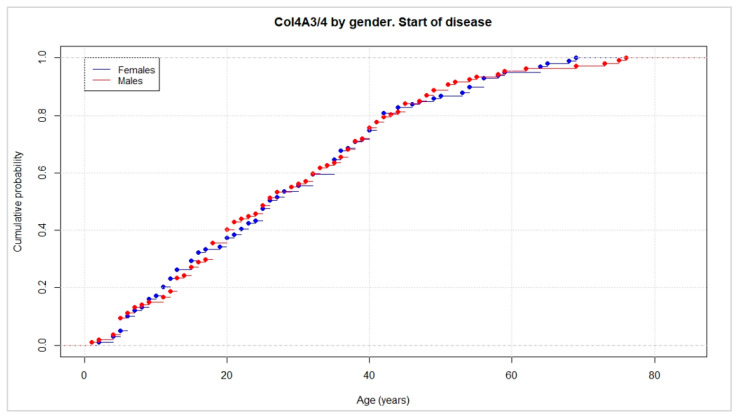
The cumulative probability of urine analytical abnormalities (microhematuria, hematuria, proteinuria) against age. This graph represents the age of onset of renal disease in the 317 patients with heterozygous deleterious mutations in COL4A3 and COL4A4 in this cohort. No differences were observed between the male (in **red**) and female (in **blue**) patients.

**Figure 2 jcm-11-04883-f002:**
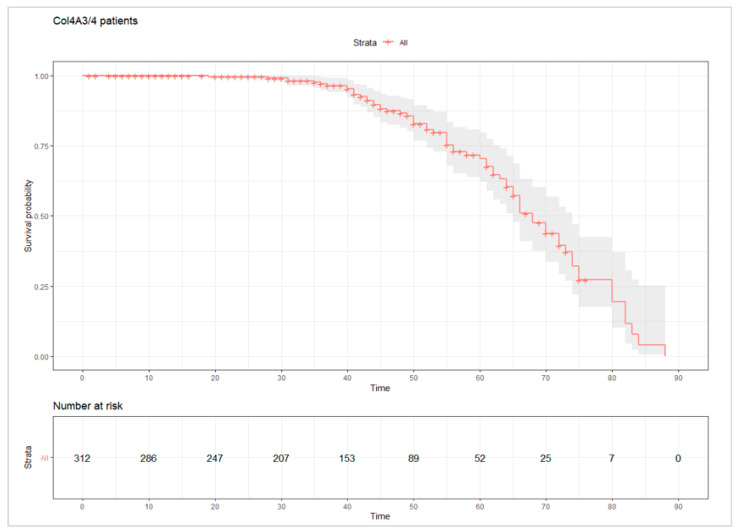
The cumulative probability of adverse renal events (ESKD, CKD, dialysis, renal transplant, or death due to renal failure). This graph represents the risk of renal failure through the occurrence of different renal events (ESKD, CKD, dialysis, renal transplant, or death due to renal failure) in patients with heterozygous deleterious mutations in the COL4A3 and COL4A4 genes. Twenty-three percent of all patients suffered some kind of renal event.

**Figure 3 jcm-11-04883-f003:**
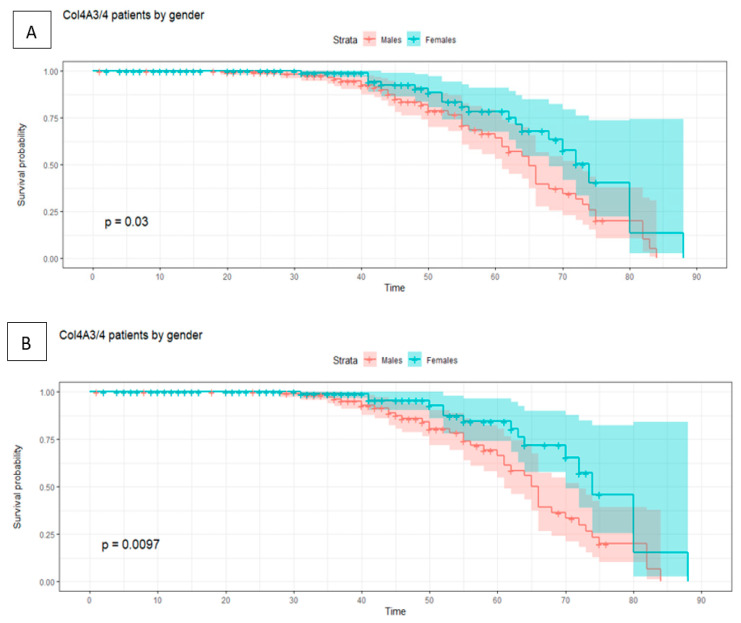
The cumulative probability of adverse renal events (ESKD, CKD, dialysis, renal transplant, or death due to renal failure) segregated by gene and sex. (**A**) This figure represents the cumulative probability of renal failure with sex segregation in patients with heterozygous deleterious mutations in COL4A3/4, observing borderline statistical differences between the males (in red) and females (in turquoise), with *p* value = 0.03. (**B**) This figure represents the cumulative probability of renal failure excluding CRD with sex segregation in patients with heterozygous deleterious mutations in COL4A3/4, observing statistical differences between the males (in red) and females (in turquoise) with a *p* value = 0.0097.

**Figure 4 jcm-11-04883-f004:**
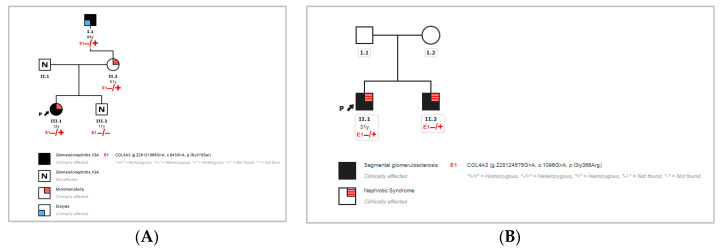
Pedigree of case 1 (**A**) and pedigree of case 2 (**B**).

**Table 1 jcm-11-04883-t001:** A summary of the clinical data on patients with heterozygous deleterious mutations in *COL4A3*.

Clinical Variables	Items	No. of Cases	Percentage (%)
**Gene**	COL4A3	136	46.70
**Mutation type**	Nonsense	0	0
	Frameshift	0	0
	Splicing	1	1.09
	Missense	90	97.83
	In-frame deletion	1	1.09
**Sex**	Male	52	53.26
	Female	45	46.74
**Presence of urine analytical alterations**	Microhematuria	18	19.57
	Hematuria	36	39.13
	Proteinuria	55	59.78
**Renal events**	ESKD	10	10.87
	CRD	4	4.35
	Dialysis	6	6.52
	Renal transplant	4	4.35
**Hearing impairment**	Hypoacusia	23	25
**Events**	**Ages (males)**	**Ages (females)**	**Total**
**Average of analytical alterations**	30	32	31
**Minimum age of onset of renal failure**	30	50	30
**Average of renal failure**	50	62	52

**Table 2 jcm-11-04883-t002:** A summary of the clinical data on patients with heterozygous deleterious mutations in *COL4A4*.

Clinical Variables	Items	No. of Cases	Percentage (%)
**Mutated gene**	COL4A4	181	53.30
**Mutation type**	Nonsense	8	7.62
	Frameshift	17	16.19
	Splicing	27	25.71
	Missense	51	48.57
	In-frame deletion	2	1.90
**Sex**	Male	55	52.38
	Female	50	47.62
**Presence of urine analytical alterations**	Microhematuria	56	53.33
	Hematuria	36	34.29
	Proteinuria	45	42.86
**Renal events**	ESKD	14	13.33
	CRD	6	5.71
	Dialysis *	3	2.86
	Renal transplant *	1	0.95
**Hearing impairment**	Hypoacusia	11	10.48
**Events**	**Age (males)**	**Age (females)**	**Total**
**Average of analytical alterations**	31	28	29
**Minimum age of onset of renal failure**	40	29	29
**Average of renal failure**	56	47	53

* These clinical data were the first event considered for the statistical analysis in these patients.

## Data Availability

Not applicable.

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
