# Peer review of "New Insights into Renal Failure in a Cohort of 317 Patients with Autosomal Dominant Forms of Alport Syndrome: Report of Two Novel Heterozygous Mutations in COL4A3"

_jcm, 2022, doi:10.3390/jcm11164883_

Round 1

Reviewer 1 Report

In the manuscript entitled, "New insights into renal insufficiency in a cohort of 317 patients with autosomal dominant forms of Alport syndrome: Report of two novel mutations in COL4A3" the authors provide descriptive information on ADAS.

Recommendations

1. More description of dataset required; these seem to be patients that were genetically screened due to known familial disease. How many were screened for this reason?  How does this effect underlying demographic characteristics and ability to extrapolate data to general populace? 

2. Appearance of disease - this needs to be defined better. In XLAS, hematuria is normally present during the first decade of life. To comment on onset of urinary abnormalities, needs clarity about how much longitudinal data available for the patients. 

3. Renal events - a better approach might include multivariate analysis with subsequent analysis of other features. 

4. New mutation - how does this tie in to the other cases? Can you access publicly available GWAS or other such databases to estimate prevalence or in AS specific cohorts?

5. By definition, most ADAS are benign and not discovered via normal clinical workup of glomerular disease since urinary abnormalities are mild do not usually prompt biopsy. Additionally genetic screening is not part of routine care at this time. Discussion should include the applicability to the ADAS population as a whole and how this relates to current clinical practice and how it might change clinical practice. 

=

Author Response

Recommendations 

  1. More description of dataset required; these seem to be patients that were genetically screened due to known familial disease. How many were screened for this reason?  How does this effect underlying demographic characteristics and ability to extrapolate data to general populace? 

Answer: We have added a clarification of this recommendation.

Line 74: For this purpose, we use a systematic search of combining key words by using PubMed resources (“COL4A3”, “COL4A4”, “mutation”, “variant”, and “Alport syndrome”, “renal insufficiency”, “renal failure”, “hematuria”, “ESKD/ESRD/CKD/CRD”) and incorporating them to an in-ternal data base. Seventy papers were revised. A total cohort of 317 patients were recruited, including four cases studied at our center who harbored two novel likely pathogenic mutations in COL4A3. The variants selected from patients reported in the screened literature were classified as pathogenic or likely pathogenic according to ACMG criteria. The allele frequency of this variants were under 0.01%, except for two variants with uncertain clini-cal significance in COL4A4 gene (p.Met1399Leu and p.Gly545Ala), which were considered low penetrant polymorphisms. For this study, we designed a specific form to collect demographic data of index cases and family members, genotype results from genetic tests, and phenotypic features, including analytical abnormalities, state of renal function, age of disease onset, and adverse events related to renal insufficiency. Twenty published reports, which matched with the search criteria and included this detailed clinical information, were gathered for the statistical analysis.

  1. Appearance of disease - this needs to be defined better. In XLAS, hematuria is normally present during the first decade of life. To comment on onset of urinary abnormalities, needs clarity about how much longitudinal data available for the patients.The revision of previous publications might introduce a bias in the longitudinal follow-up of the patients.

Answer: We agree with the reviewer in the limitations of this study

Line 324: Although there is a relative high incidence of microhematuria and normal renal function in carriers of deleterious mutations in COL4A3/4 [39],

  1. Renal events - a better approach might include multivariate analysis with subsequent analysis of other features.  

Answer: Further studies could be applied later.

Previous publications have included multivariate analysis identifying associations with factors like the presence proteinuria. Our internal statistical studies did not allow us to detect relevant conclusions in this aspect. The increase of the cohort could be allow us to perform robust multivariate analysis.

  1. New mutation - how does this tie in to the other cases? Can you access publicly available GWAS or other such databases to estimate prevalence or in AS specific cohorts?

Answer: clarification

Line 240: (Gly315Ser) This variant was absent in gnomAD database.

Line 256 (Gly366Arg) being oberved in three heterozygous carriers in gnomAD database.

Line 353: The presence of heterozygous carriers of variants in COL4A3/4 in general population would be explained by the subclinical forms of ADAS in patients who harbor pathogenic mutations in such genes.

  1. By definition, most ADAS are benign and not discovered via normal clinical workup of glomerular disease since urinary abnormalities are mild do not usually prompt biopsy. Additionally genetic screening is not part of routine care at this time. Discussion should include the applicability to the ADAS population as a whole and how this relates to current clinical practice and how it might change clinical practice. 

Answer: Update information

Line 359: These results are important for the stratification of patients with these features in order to improve the management of autosomal dominant AS and better understanding the factors which influence the inter- and intra-familial variability of ADAS.

Reviewer 2 Report

Garcia-Aznar et al. submit a paper composed of 2 parts, both devoted to Autosomal Dominant Alport syndrome (ADAS)

a)      They claim to review the published cases of ADAS, and present a cumulative probability of abnormalities of urinalysis  as well as of adverse kidney outcomes (dialysis, TP, ESKD,…), both as a function of age and gender.

b)     They report 2 new families with novel mutations in COL4A3

General comments

1.      Both parts of the paper suffer from major problems. The key one is the complete absence of any definition of ADAS. If this reviewer understood correctly, all papers with ADAS in the title or text were included in part A? But such a review definitely should have predefined inclusion criteria , and subsequently applied them to published papers. Was the analysis of all 3 COL4A genes (3,4,5) mandatory for inclusion? By which tests? And at what date (genetic tests have progressed enormously in recent years)

2.      In addition, regarding part A, the authors should clarify whether their review was systematic (and again , if so, which criteria and key words, which databases etc…, this was probably not a systematic review) or rather narrative.

3.      The authors should further clarify how the kidney outcomes were defined in their review. Did they use the events reported by the authors of each paper , or did they check for the accuracy of the reported outcomes and their definition (albuminuria, hematuria,etc…). Similarly , how was hypoacousia defined? Clinical? Auditory testing? Threshold for meeting “hypoacousia”?

4.      How was missing data handled for part A. Were patients censored at the time of last follow-up?

5.      Again, regarding part A, the authors should include in their discussion a section on the high probability of reporting bias (more probability of reporting of severe cases)

6.      Many references of the paper are outdated and should be replaced by the most current ones. In particular the authors should quote the 2022 guidelines for genetic testing and management by J Savige et al. (CJASN 2022) , and the recent cohort by R Torra’s group (Furlano et al AJKD 2021)

7.      Regarding part B, the description of cases is at best very imperfect, and thus generates more questions than answers. The urinalysis of case 1 should be described in detail and eGFR provided. The authors should  mention whether the grandfather on dialysis since age of 70 is the father of the mother of the propositus of the other grandfather! Regarding case2, some details about the nephrotic syndrome should be provided. Was this a clinical nephrotic syndrome? Albumin level? Was the onset of disease progressive or sudden? Etc…

8.      The authors should use the widely accepted KDIGO nomenclature and thus prefer ESKD or kidney failure , rather than ESKD, avoid “renal insufficiency” etc…(see  Nomenclature for kidney function and disease: report of a Kidney Disease: Improving Global Outcomes (KDIGO) Consensus Conference. Levey AS, Eckardt KU, Dorman NM, Christiansen SL, Hoorn EJ, Ingelfinger JR, Inker LA, Levin A, Mehrotra R, Palevsky PM, Perazella MA, Tong A, Allison SJ, Bockenhauer D, Briggs JP, Bromberg JS, Davenport A, Feldman HI, Fouque D, Gansevoort RT, Gill JS, Greene EL, Hemmelgarn BR, Kretzler M, Lambie M, Lane PH, Laycock J, Leventhal SE, Mittelman M, Morrissey P, Ostermann M, Rees L, Ronco P, Schaefer F, St Clair Russell J, Vinck C, Walsh SB, Weiner DE, Cheung M, Jadoul M, Winkelmayer WC.)

Kidney Int. 2020 PMID: 32409237

Author Response

General comments 

  1. Both parts of the paper suffer from major problems. The key one is the complete absence of any definition of ADAS. If this reviewer understood correctly, all papers with ADAS in the title or text were included in part A? But such a review definitely should have predefined inclusion criteria , and subsequently applied them to published papers. Was the analysis of all 3 COL4A genes (3,4,5) mandatory for inclusion? By which tests? And at what date (genetic tests have progressed enormously in recent years) 

Answer: Updating information

Line 306: The recent advances in NGS have enabled genetic testing to be performed for the diagnosis of AS as first-line diagnosis, allowing to analyze single variants and copy number varia-tion. Genotyping of family members in a context of AS has revealed a wide spectrum of the phenotype, including genotype-phenotype correlations. Dominant forms of AS repre-sents up to 30% of AS suspicions [16, 26] and are characterized by a spectrum of pheno-types ranging from progressive kidney disease with extrarenal abnormalities to isolated hematuria with a nonprogressive or very slowly course with wide range of expression, inheritance and penetrance.

  1. In addition, regarding part A, the authors should clarify whether their review was systematic (and again, if so, which criteria and key words, which databases etc…, this was probably not a systematic review) or rather narrative. 

Answer: We add a clarification of this recommendation.

Line 74: For this purpose, we use a systematic search of combining key words by using PubMed resources (“COL4A3”, “COL4A4”, “mutation”, “variant”, and “Alport syndrome”, “renal insufficiency”, “renal failure”, “hematuria”, “ESKD/ESRD/CKD/CRD”) and incorporating them to an in-ternal data base. Seventy papers were revised. A total cohort of 317 patients were recruited, including four cases studied at our center who harbored two novel likely pathogenic mutations in COL4A3. The variants selected from patients reported in the screened literature were classified as pathogenic or likely pathogenic according to ACMG criteria. The allele frequency of this variants were under 0.01%, except for two variants with uncertain clini-cal significance in COL4A4 gene (p.Met1399Leu and p.Gly545Ala), which were considered low penetrant polymorphisms. For this study, we designed a specific form to collect demographic data of index cases and family members, genotype results from genetic tests, and phenotypic features, including analytical abnormalities, state of renal function, age of disease onset, and adverse events related to renal insufficiency. Twenty published reports, which matched with the search criteria and included this detailed clinical information, were gathered for the statistical analysis.

  1. The authors should further clarify how the kidney outcomes were defined in their review. Did they use the events reported by the authors of each paper , or did they check for the accuracy of the reported outcomes and their definition (albuminuria, hematuria,etc…). Similarly, how was hypoacousia defined? Clinical? Auditory testing? Threshold for meeting “hypoacousia”? 

Answer: All events and clinical features including the term of hypoacusia, which were used as data for the analysis were explicitly mentioned in the publications avoiding bias of absent information.

  1. How was missing data handled for part A. Were patients censored at the time of last follow-up? 

Answer: The patients included in the Kaplan-Mayer analysis were censored at the time of last follow-up.

Line 63: The differences in the longitudinal follow-up of each patient belonging from the publications is a technical limitation of this analysis.

  1. Again, regarding part A, the authors should include in their discussion a section on the high probability of reporting bias (more probability of reporting of severe cases) .

Answer: We add a clarification of this recommendation.

Line 346: The statistical analysis of this cohort, specially from the revision of the literature, may be subject to influence from severe case reports, published because of the striking phenotype of the patients.

  1. Many references of the paper are outdated and should be replaced by the most current ones. In particular the authors should quote the 2022 guidelines for genetic testing and management by J Savige et al. (CJASN 2022), and the recent cohort by R Torra’s group (Furlano et al AJKD 2021)

Answer: Corrected

  1. Regarding part B, the description of cases is at best very imperfect, and thus generates more questions than answers. The urinalysis of case 1 should be described in detail and eGFR provided. The authors should mention whether the grandfather on dialysis since age of 70 is the father of the mother of the propositus of the other grandfather! Regarding case2, some details about the nephrotic syndrome should be provided. Was this a clinical nephrotic syndrome? Albumin level? Was the onset of disease progressive or sudden? Etc… 

Answer: Updated clinical information.

Line 95: Case 1: A 15-year-old girl, that presented at the age of 13 with transient gross hema-turia and massive proteinuria, with normal serum electrolytes, albumin, urea and creati-nine values (albumin, 3.2 g/dL; urea, 24 mg/dL; serum creatinine, 0.6 mg/dL; and esti-mated glomerular filtration rate [eGFR], 106.7 mL/min/1.73). She has also normal hemo-globin level (12.8 mg/dL), normal white blood cell count (9.3 x 10^3/µL), and normal platelet count (240 x 10^9/L). Peripheral blood film showed anisopoikilocytes (irregularly shaped and burr cells), but no leukocyte inclusion bodies. She has no history of hearing difficulties nor vision problems, but her mother (43-year-old) has history of hearing diffi-culties, microscopic hematuria and history of transient gross hematuria attributed to uri-nary tract infections, being the eGFR of 80.0 mL/min/1.73. Her grandfather (father of the proband’s mother) had renal failure and was under dialysis at the age of 70.  On exami-nation following referral, the patient had no edema, normal blood pressure (93/63 mmHg, at the 50th percentile for age and height), and normal heart rate. Urine examination re-vealed mild proteinuria; albumin-to-creatinine ratio [ACR], 140 to 286 mg/g; and hematu-ria; being serum electrolyte, urea and creatinine normal, and the eGFR, 96.2 mL/min/1.73. Ultrasonography revealed a normal-sized kidneys with normal parenchymal echogenici-ty with some loss of corticomedullary differentiation. Slit-lamp examination and audiom-etry were normal. She is on regular follow-up, and her proteinuria is being treated with an angiotensin-converting enzyme inhibitor (ACEI). There was significant reduction in pro-teinuria that ranges from normal to mildly to low-moderate increased (ACR, 24 to 100 mg/g), but persistent microscopic hematuria. She did not present new episodes of gross hematuria.

Line 117: Case 2: This patient is a 31-year-old male that start at the age of 26 years with corti-coresistant nephrotic syndrome (edema, hypoalbuminemia, and proteinuria with >3.5g/day). The renal biopsy revealed lesions compatible with a diagnosis of focal and segmental hyalinosis. The immunosuppressor treatment was initiated with cyclosporine and mycophenolate, showing bad response. During next years of follow-up, he has suf-fered slow deterioration of the renal function maintaining high levels of proteinuria. Cur-rently, he is currently 31 years old and develops ESKD. A brother of this proband has the same condition.

  1. The authors should use the widely accepted KDIGO nomenclature and thus prefer ESKD or kidney failure , rather than ESKD, avoid “renal insufficiency” etc…(see  Nomenclature for kidney function and disease: report of a Kidney Disease: Improving Global Outcomes (KDIGO) Consensus Conference. Levey AS, Eckardt KU, Dorman NM, Christiansen SL, Hoorn EJ, Ingelfinger JR, Inker LA, Levin A, Mehrotra R, Palevsky PM, Perazella MA, Tong A, Allison SJ, Bockenhauer D, Briggs JP, Bromberg JS, Davenport A, Feldman HI, Fouque D, Gansevoort RT, Gill JS, Greene EL, Hemmelgarn BR, Kretzler M, Lambie M, Lane PH, Laycock J, Leventhal SE, Mittelman M, Morrissey P, Ostermann M, Rees L, Ronco P, Schaefer F, St Clair Russell J, Vinck C, Walsh SB, Weiner DE, Cheung M, Jadoul M, Winkelmayer WC.) 

Answer: we agree with the reviewer in correct the nomenclature base on KDIGO recommendations.

Round 2

Reviewer 2 Report

The revised version of the paper submitted by Garcia-Aznar et al is somewhat improved.
Yet, this reviewer is still concerned by multiple aspects of the revised draft, as detailed below

General comments

1.      As pointed out in my initial review, pre-defining criteria for Autosomal Dominant (AD) Alport Syndrome (AS) is critical, before reviewing the relevant literature. The authors now provide the keywords of their pubmed search, as well as the criteria for accepting variants as probably or definitely pathogenic (these criteria are international ones). But to my surprise, and deep disappointment, they do not clarify which criteria were required to meet the AD title! NOWHERE in the revised section, they clarify that relevant variants should be heterozygous (which should implicitly from the title be the case) Such a key point should be explicit! Was there a requirement for a family history? Were subjects with neomutations excluded or included? How was a relevant family history defined (on a clinical basis? With demonstration of transmission of the pathogenic variant within the family?) etc…

2.      Secondly, the demonstration that ESKD is reached some years earlier in males than females is of potential interest. Yet the take home message in the abstract is all except clearcut: “ suggesting that males begin  experiencing faster deterioration of renal function earlier than women “. My concern is related to the coexistence of faster and earlier in the same sentence. Do the authors mean faster ? because regarding earlier , figure 1 shows that urine abnormalities are detected at a similar age in males and females.

Specific comments

1.      Table 2 mentions 14 cases of ESKD, 3 cases with dialysis and 1 kidney TP. Are the 3 dialysis and single kidney TP cases included in the 14 ESKD cases. Please clarify!

2.      The actual significance of lines 378-380 is unclear to this reviewer : “Although there is a relative high incidence of microhematuria and normal renal function 378 in carriers of deleterious mutations in COL4A3/4 , the revision of previous publica tions might introduce a bias in the longitudinal follow-up of the patients” What does this mean

3.      At lines 399-401, the authors should consider introducing the concept of publication bias of more severe cases

4.      Reference 10 includes 2 different refs and should be cleaned!

Author Response

General comments

  1. As pointed out in my initial review, pre-defining criteria for Autosomal Dominant (AD) Alport Syndrome (AS) is critical, before reviewing the relevant literature. The authors now provide the keywords of their pubmed search, as well as the criteria for accepting variants as probably or definitely pathogenic (these criteria are international ones). But to my surprise, and deep disappointment, they do not clarify which criteria were required to meet the AD title! NOWHERE in the revised section, they clarify that relevant variants should be heterozygous (which should implicitly from the title be the case) Such a key point should be explicit! Was there a requirement for a family history? Were subjects with neomutations excluded or included? How was a relevant family history defined (on a clinical basis? With demonstration of transmission of the pathogenic variant within the family?) etc…

Reply: We fully agree with this reviewer in incising with this aspect and clarifying this question.

Line 64: Therefore, the high variability of manifestations in ADAS and presence of undiag-nosed cases make difficult stablishing the clinical diagnosis before a genetic test is per-formed. Inclusion criteria in previous cohorts of ADAS have included familial with auto-somal dominant inheritance microhematuria and proteinuria with or without CKD or abnormalities in the glomerular basement membrane [9]. Noteworthy, the family history may be not a necessary criterion since a few of cases are de novo.

Title: We have explicitly included “heterozygous in the title”: New insights into renal failure in a cohort of 317 patients with autosomal dominant forms of Alport syndrome: Report of two novel heterozygous mutations in COL4A3.

Key words: We have modified the kew words in other to be consistent with the pubmed search of the article. Alport syndrome; Autosomal dominant inheritance; ESKD; COL4A3; COL4A4.

Line174: This cohort included patients previously reported in the scientific literature (313) and four cases studied at our center who harbored pathogenic or likely pathogenic heterozygous mutations in COL4A3 or COL4A4.

Regarding the literature review, the presence of monoallelic disease-causing mutation was the strongest criterium to consider if the mutation is associated with autosomal dominant forms since, the clinical spectrum in such cases may vary and subclinical findings can lead to undiagnosed cases. De novo cases can also be present. In the internal probands, the positive family history in the first case was considered according to the intrafamilial variability matching with ADAS. In the second case, the lack of a second disease-causing mutation in COL4A3 gene and the impossibility to study the parents suggest an ADAS phenotype.

  1. Secondly, the demonstration that ESKD is reached some years earlier in males than females is of potential interest. Yet the take home message in the abstract is all except clearcut: “ suggesting that males begin  experiencing fasterdeterioration of renal function earlier than women “. My concern is related to the coexistence of faster and earlier in the same sentence. Do the authors mean faster ? because regarding earlier , figure 1 shows that urine abnormalities are detected at a similar age in males and females.

Reply: We fully agree with this reviewer in this clarification. We have removed the consideration of faster deterioration because of the lack of demonstrating data supporting this assumption.

Specific comments

  1. Table 2 mentions 14 cases of ESKD, 3 cases with dialysis and 1 kidney TP. Are the 3 dialysis and single kidney TP cases included in the 14 ESKD cases. Please clarify!

Reply: The renal event which was taken in the Kaplan-Meier curve was the first one that occurred. Thus, the cases with dialysis or TP were the unique and first event counted in the analysis.

Line 185: *These clinical data were the first event considered for the statistical analysis in those patients.

  1. The actual significance of lines 378-380 is unclear to this reviewer : “Although there is a relative high incidence of microhematuria and normal renal function 378 in carriers of deleterious mutations in COL4A3/4 , the revision of previous publica tions might introduce a bias in the longitudinal follow-up of the patients” What does this mean

Reply: We agree with the reviewer in this confusing sentence.

Line 350: Correction.

  1. At lines 399-401, the authors should consider introducing the concept of publication bias of more severe cases

Reply: We agree with the reviewer in this confusing sentence.

Line 348: Correction.

  1. Reference 10 includes 2 different refs and should be cleaned!

Reply: The second reference has been deleted.
